# New Cardenolides from Biotransformation of Gitoxigenin by the Endophytic Fungus *Alternaria eureka* 1E1BL1: Characterization and Cytotoxic Activities

**DOI:** 10.3390/molecules26103030

**Published:** 2021-05-19

**Authors:** Erdal Bedir, Çiğdem Karakoyun, Gamze Doğan, Gülten Kuru, Melis Küçüksolak, Hasan Yusufoğlu

**Affiliations:** 1Department of Bioengineering, Faculty of Engineering, İzmir Institute of Technology, 35430 Urla-İzmir, Turkey; gamzedogan@iyte.edu.tr (G.D.); gultenkuru@iyte.edu.tr (G.K.); meliskucuksolak@iyte.edu.tr (M.K.); 2Department of Pharmacognosy, Faculty of Pharmacy, Ege University, 35100 Bornova-İzmir, Turkey; 3Department of Pharmacognosy, College of Pharmacy, Prince Sattam Bin Abdulaziz University, Al-Kharj 11942, Saudi Arabia; h.yusufoglu@psau.edu.sa

**Keywords:** *Nerium oleander* L., cardenolides, oleandrin, gitoxigenin, biotransformation, endophytic fungus, *Alternaria eureka* 1E1BL1, cytotoxicity

## Abstract

Microbial biotransformation is an important tool in drug discovery and for metabolism studies. To expand our bioactive natural product library via modification and to identify possible mammalian metabolites, a cytotoxic cardenolide (gitoxigenin) was biotransformed using the endophytic fungus *Alternaria eureka* 1E1BL1. Initially, oleandrin was isolated from the dried leaves of *Nerium oleander* L. and subjected to an acid-catalysed hydrolysis to obtain the substrate gitoxigenin (yield; ~25%). After 21 days of incubation, five new cardenolides **1**, **3**, **4**, **6**, and **8** and three previously- identified compounds **2**, **5** and **7** were isolated using chromatographic methods. Structural elucidations were accomplished through 1D/2D NMR, HR-ESI-MS and FT-IR analysis. *A. eureka* catalyzed oxygenation, oxidation, epimerization and dimethyl acetal formation reactions on the substrate. Cytotoxicity of the metabolites were evaluated using MTT cell viability method, whereas doxorubicin and oleandrin were used as positive controls. Biotransformation products displayed less cytotoxicity than the substrate. The new metabolite **8** exhibited the highest activity with IC_50_ values of 8.25, 1.95 and 3.4 µM against A549, PANC-1 and MIA PaCa-2 cells, respectively, without causing toxicity on healthy cell lines (MRC-5 and HEK-293) up to concentration of 10 µM. Our results suggest that *A. eureka* is an effective biocatalyst for modifying cardenolide-type secondary metabolites.

## 1. Introduction

Microbial biotransformation is one of the most preferred and powerful procedures in natural product drug discovery and development studies enabling specific biochemical modifications of substrate molecules [1]. An enzyme, a living cell, or an inactivated microorganism containing several specific enzymes can be used for biotransformation processes leading to the production of new drug candidates with better properties. Microbial enzyme systems are able to catalyze regio- and stereo-specific reactions which are challenging issues in chemical synthesis [2,3,4]. These enzymes are to employ diverse chemical reactions such as oxygenation, oxidation/reduction, cyclization, epoxidation, dehydrogenation, acetylation/deacetylation and epimerization. Besides, most favored metabolic products of the given substrate can be predicted especially via fungal biotransformation [5]. Thus, foreseeing metabolic fate of drug candidates provides insight into mammalian drug metabolism at preclinical stages. Microbial transformation, as a modern, time- and cost-efficient molecular modification approach, is of increasing interest to pharmaceutical industries. Also, the use of microbial biotransformation has become one of the essential parts of white biotechnology and green chemistry movements due to requiring less energy and creating less waste [6].

*Nerium oleander* L. is a perennial evergreen shrub that belongs to the family Apocynaceae. It is a widely cultivated ornamental herb, native to the Mediterranean region. In ethnobotanical literature, *N. oleander* is reported as a traditional medicine being used as abortifacient and for the treatment of a wide range of diseases such as dermatitis, eczema, skin cancer, viral infections, asthma, epilepsy, malaria, tumors, corns, leprosy, ringworm, and warts [7]. All parts of this plant are known to be poisonous to human beings, animals, and some insects. Both toxic effects and bioactivities of *N. oleander* are mainly attributed to the cardiac glycoside content of the plant [8,9].

Recently, *N. oleander* has attracted a great deal of attention due to promising anticancer activity of two patented water extracts: Anvirzel^®^ and Breastin^®^ [10,11]. Anvirzel^®^, a sterile hot aqueous extract of *N. oleander*, has been recognized as a potent anticancer agent against advanced solid tumors. A phase I clinical trial study demonstrated the safety of intramuscular administration of Anvirzel^®^ up to the dose of 1.2 mL/m^2^/day. Other clinically tested anticancer natural product, Breastin^®^, is the cold-water extract of *N. oleander* exhibiting selective antitumor activities against brain tumor SF268, lung carcinoma LXF1221L, pancreas carcinoma PANC-1 and prostate cancer DU 145. Phytochemical studies have demonstrated that potent anticancer activity of the water extract of *N. oleander* mostly arises from its monoglycosidic cardenolide content such as oleandrin, odoroside A, odoroside H, and neritaloside [11]. In particular, Anvirzel^®^ and Breastin^®^ contain considerably high amounts of oleandrin, which is regarded as the principal active constituent anticancer [12].

Oleandrin, possesses a cardenolide-type aglycone called oleandrigenin together with a dideoxy sugar moiety called oleandrose (Figure 1). The main physiological action of cardenolides is cardiotonic effect and depends on disrupting sodium potassium pump (Na^+^/K^+^-ATPase) functions leading to increased levels of intracellular calcium ions (Ca^++^). Thus, cardenolide-type compounds generate positive inotropic and negative chronotropic action on heart muscles, and consistently improve the prognosis of patients with congestive heart failure. In addition to anticancer and heart contractility promoting effects, oleandrin is also known to possess anti-oxidant, diuretic, anti-inflammatory, neuroprotective and anti-viral activities [13,14,15]. Furthermore, a recent study has reported that oleandrin is a potent anti-viral agent providing both prophylactic and curative treatment for COVID-19. According to in vitro experiments, administration of oleandrin at 0.05 and 0.1 µg/mL doses resulted in 78- and 100-fold reduction in infectious virus production, respectively [16].

Gitoxigenin is desacetyl congener of oleandrigenin (Figure 1), of which the concentration in biological samples of patients is commonly monitored as one of the follow-up markers in cases of oleander poisoning [17]. Despite the fact that high concentrations of gitoxigenin have poisonous effects, it has been known that certain doses have therapeutic effects on congestive heart failure and atrial arrhythmia. Thus, gitoxigenin is a promising natural product present in a variety of plants [18,19].

Based on our latest literature search, there are only a few studies on biological activity of gitoxigenin [20,21,22,23,24]. Low et al. screened 502 natural products for anti-viral activity against dengue virus via immunofluorescence assay with the hit criteria of ≥40% inhibition, and gitoxigenin was detected as one of the 30 hit compounds exhibiting 49.01% inhibition [25]. Xie et al. conducted a high throughput screening study on more than 3000 compounds and reported potent anticancer activity of gitoxigenin against human colon cancer cell line [26]. Milutinovic et al. demonstrated the anticancer effect of gitoxigenin against HeLa cells through a homogeneous plate-reader assay (EC_50_: 2.1 µM) [17]. Also, cytotoxic activity of gitoxigenin against three cancerous cell lines (human renal adenocarcinoma, TK-10; human breast adenocarcinoma, MCF-7; and human melanoma, UACC-62) was investigated via sulphorhodamine B colorimetric method in a previous work by López-Lázaro et al., and gitoxigenin was reported as potent cytotoxic agent showing IC_50_ values in the range of 0.415 to 2.83 µM [21].

Despite the promising potential of cardiac glycosides as drug candidates, their use in medicinal applications is limited because of their aggressive pharmacokinetic and toxicologic behaviours. Thus, there is an urgent need for research on drug development studies on cardenolide glycosides and their aglycones, in order to overcome these challenges. For this purpose, biotransformation might be a useful tool for investigation of structure-activity relationships (SAR), prediction of drug metabolism and discovery of new drug candidates with increased bioavailability. However, only a few studies have been conducted on microbial biotransformation of gitoxigenin [27,28].

Recently our group have performed molecular modification studies on bioactive natural products and reported interesting findings [29,30,31,32]. Taking into consideration of these findings and our preliminary screening studies, *A. eureka* 1E1BL1 was proven to have high transformation capacity on steroidal nuclei. Therefore, within the first part of our current work on the constituents of *Nerium oleander* L., the cardiac glycoside oleandrin was transformed using the same endophytic fungus in addition to *Phaeosphaeria* sp. 1E4CS-1, which resulted in isolation of five new oleandrin derivatives with significant cytotoxic activities [33].

In the present study, we aimed to perform a microbial biotransformation study on gitoxigenin, which was obtained by acid-catalyzed hydrolysis of oleandrin. Incubation of gitoxigenin with *A. eureka* 1E1BL1 has led to the production of five previously undescribed cardenolides together with three known metabolites. Microbial oxygenations at C-7 and C-12 sites were observed while dimethyl acetal formation on gitoxigenin core structure was reported here for the first time. All isolates and the substrate were evaluated for cytotoxic activity against several cancer cell lines (PANC-1, MIA PaCa-2, DU 145 and A549) and two healthy cell lines (MRC-5 and HEK-293) serving doxorubicin and oleandrin as positive controls.

## 2. Results and Discussion

### 2.1. Isolation and Purification of Oleandrin from N. oleander L. and Preparation of Substrate Gitoxigenin and Obtaining Biotransformation Products

Oleandrin (4.0 g) was isolated from the dried leaves of *N. oleander* L. and subjected to an acid-catalyzed hydrolysis affording gitoxigenin (25% yield) (Figure 1). The structural identification of gitoxigenin was established unambiguously by spectral methods (see Section 3.6 and Appendix A). A twenty-one-day biotransformation experiment of gitoxigenin by *Alternaria eureka* 1E1BL1 yielded eight metabolites (Figure 2).

### 2.2. Structural Elucidations of Biotransformation Products

The ^1^H and ^13^C NMR data of the new metabolites **1**, **3**, **4**, **6** and **8** are presented in Table 1 and Table 2, respectively. Furthermore, the NMR data of **2** and **7**, which were tentatively identified in the previous studies, are herein reported for the first time (Table 1 and Table 2).

The molecular formula of **1** was determined as C_23_H_32_O_6_ based on the major ion peak at *m*/*z* 403.2123 [M-H]^−^ (calcd. 403.2126 for C_23_H_32_O_6_ [M-H]^−^). The characteristic hydroxymethine proton at C-3 was not present in the ^1^H-NMR spectrum. Besides, the downfield shifts of C-2 and C-4 (ca. 8.3 ppm and 8.7 ppm) compared to those of gitoxigenin implied an alteration on the ring A. Accordingly, in the ^13^C-NMR spectrum, the carbonyl signal observed at δ_C_ 212.7 was readily assigned to C-3, which was substantiated by the observation of HMBC cross-peaks from C-3 (δ_C_ 212.7) to one of H_2_-2 (δ_H_ 2.21, td, *J* = 14.4; 5.3 Hz) and H_2_-4 (δ_H_ 2.43, m) protons. Apart from the characteristic low-field signals observed, an additional oxymethine signal at δ_H_ 3.84 was apparent, corresponding to a carbon signal at δ_C_ 69.1 in the HSQC spectrum. When the COSY spectrum was inspected starting from the H_2_-4 resonances (δ_H_ 2.10 and 2.43), the spin system to the new resonance at δ_H_ 3.84 was tracked clearly [H_2_-4/H-5 (δ_H_ 1.86, m)/H_2_-6 (δ_H_ 1.50 and 1.83 m)/H-7 (δ_H_ 3.84, td, *J* = 10.6, 4.9 Hz)], suggested H-7 to be an oxymethine proton. The significant downfield shifts of C-6 (δ_C_ 35.9) and C-8 (δ_C_ 45.6) (ca. 8.2 and 2.8 ppm compared to those of gitoxigenin, respectively), and the long-distance correlations in the HMBC spectrum from C-7 (δ 69.1) to H_2_-6 (δ_H_ 1.50 m; 1.83 m) and H-8 (δ_H_ 1.69, dd, *J*=11.8; 10.4 Hz) verified the monooxygenation at C-7 position (Figure 3).

The relative orientation of the hydroxy group was determined using 2D-NOESY. The H-7 (δ_H_ 3.84) signal showed NOESY correlation with the *α*-oriented H-9 (δ_H_ 1.58 m) establishing the *β*-orientation of the hydroxy group at C-7. Consequently, the structure of compound **1** was characterized as 7*β*-hydroxy-3-oxogitoxigenin.

The molecular formula of **2** was determined as C_23_H_32_O_5_ based on the HR-ESI-MS analysis (*m*/*z* 433.2227 [M + FA-H]^−^, calcd. 433.2231 for C_23_H_32_O_5_) with eight indices of hydrogen deficiency (HD). Apart from the absence of hydroxymethine signal of C-3 and the downfield shift of carbon resonances belonging to A-ring, the characteristic signals of the gitoxigenin were present for **2**. In the ^13^C-NMR spectrum, the resonance at δ_C_ 212.7 was evident for modification of C-3 secondary alcohol to a ketone, substantiating the increase in HD. Moreover, the HMBC correlations of C-3 with one of H_2_-2 (δ_H_ 2.32, dd, *J* = 14.5; 6.4 Hz) and H_2_-4 (δ_H_ 2.62 m) resonances, justified the oxidation at C-3 (Figure 4). Consequently, compound **2** was established as 3-oxo-gitoxigenin, a previously identified compound [34,35].

The HR-ESI-MS spectrum of **3** showed a major ion peak at *m*/*z* 419.2075 (calcd. 419.2075 for C_23_H_32_O_7_ [M-H]^−^) indicating a molecular formula of C_23_H_32_O_7_. The disappearance of the oxymethine signal of C-3 and the presence of a carbonyl carbon signal at δ_C_ 214.5 implied a similar modification, as in **1** and **2**. Examination of the HMBC spectra proved oxidation at C-3. In the spectrum, two additional low-field resonances (δ_H_ 3.91 and δ_H_ 3.36) were observed. Besides, two extra down-field carbon signals at δ 70.3 and δ 75.9 were also noted. In order to deduce the oxygenation positions, the 2D NMR spectra were studied thoroughly. Like **1**, a monooxygenation at C-7 was confirmed via inspection of the COSY spectrum (H-7: δ_H_ 3.91). The second hydroxy group was located at C-12 on the basis of the HMBC spectrum, which showed a cross-peak between the C-12 (δ_C_ 75.9) signal and H-18 (δ_H_ 0.89 s). The relative orientations of hydroxy groups at C-7 and C-12 were determined based on the 2D-NOESY data. The H-12 (δ_H_ 3.36, dd *J* = 11.8, 4.1 Hz) signal showed NOE correlation with α-oriented H-17 (δ_H_ 3.58, d, *J* = 7.6 Hz) verifying the orientation of C-12 hydroxy as *β*. Likewise, based on the NOE correlation observed between H-7 (δ_H_ 3.91, td *J* = 11.6; 11.2; 4.7 Hz) and *α*-oriented H-9 (δ_H_ 1.88, m), C-7 hydroxy group was deduced as *β*-oriented. Finally, the structure of **3** was assigned as 7*β*,12*β*-dihdyroxy-3-oxo-gitoxigenin.

The molecular formula of **4** was found to be C_25_H_38_O_7_ based on the major ion peak at *m*/*z* 449.2541 [M-H]^−^ (calcd. 449.2545 for C_25_H_38_O_7_ [M-H]^−^). Inspection of the ^1^H-NMR spectrum of **4** showed two *O*-methyl groups (δ_H_ 3.12 and δ_H_ 3.19) corresponding to the same carbon resonance (δ_C_ 47.7) in the HSQC spectrum suggesting a geminal dimethoxy substitution. Based on the absence of a low-field oxymethine signal due to H-3 in the ^1^H-NMR spectrum, and the observation of a new quaternary signal (δ_C_ 100.4) displaying long-range correlations with δ_H_ 3.12 and δ_H_ 3.19 signals in the HMBC spectrum, a geminal dimethoxy substitution at C-3 was deduced unambiguously. An additional low-field proton signal was also present at δ_H_ 3.96 indicating an oxymethine group in the structure. Examination of the COSY and HSQC spectra suggested the monooxygenation position to be C-7, as in **3**. This assumption was confirmed by the HMBC experiment, which showed the long-range correlations from H_2_-6 (δ_H_ 1.57 and 1.92) and H-8 (δ_H_ 1.72) to the 70.9 ppm resonance (C-7). The stereochemistry of the hydroxy group at C-7 was established via 2D-NOESY spectrum. The observed cross peaks from H-7 (δ_H_ 3.96, m) to *α*-oriented protons H-9 (δ_H_ 1.54, m) and H-17 (δ_H_ 2.91, dd, *J* = 7.0; 3.0 Hz) substantiated the orientation of C-7 hydroxy as *β*. Based on this evidence, the structure of **4** was determined as 3-dimethylacetal-7*β*-hydroxygitoxigenin.

Compound **5** was identified as 3-epigitoxigenin by comparison of the spectroscopic data with those previously reported by Kawaguchi et al. [36].

The major peak at *m*/*z* 405.2280 [M-H]^−^ (calcd. 405.2283 for C_23_H_34_O_6_ [M-H]^−^) in the HR-ESI-MS spectrum of compound **6** indicated 16 amu increase over gitoxigenin and **5** (3-epi-gitoxigenin), implying a hydroxylated analogue of either of them. The ^1^H- and ^13^C-NMR spectra of **6** were found to be very similar to those of **5**, except for an additional proton at δ_H_ 4.26 corresponding to a carbon signal at δ_C_ 70.5 in the HSQC spectrum (Table 1 and Table 2). In the COSY spectrum, the spin system starting from the H-8 (δ_H_ 1.95, m)/H-7 (δ_H_ 4.26, m)/H_2_-6 (δ_H_ 1.80, m and 2.14 td, *J* = 12.2, 4.9 Hz), together with the long-range correlations from C-7 (δ_C_ 70.5) to H_2_-6 and H-8 in the HMBC spectrum (see Figure 3) helped us to determine the location of hydroxylation as C-7. The relative configuration of C-7 was clarified via the interpretation of 2D-NOESY spectrum which indicated a correlation between *α*-oriented H-16 (δ_H_ 4.94, m) and one of H-15 protons appeared at δ_H_ 2.67. In turn, the cross peak from α-oriented H-15 (δ_H_ 2.67, m) to H-7 (δ_H_ 4.26, m) confirmed the orientation of hydroxy group on C-7 as *β*. Additionally, the cross peak from H-3 (δ_H_ 3.85, m) to *β*-oriented H-5 (δ_H_ 1.51, m) in the 2D-NOESY spectrum was evident to verify the epimerization at C-3. Consequently, the structure of **6** was elucidated as 7*β*-hydroxy-3-epi-gitoxigenin.

In the HR-ESI-MS of **7**, the major ion peak was observed at *m*/*z* 449.2179 [M + FA-H]^−^ (calcd. 449.2181 for C_23_H_32_O_6_ [M + FA-H]^−^), indicating the same molecular weight with that of **1**. The presence of characteristic carbonyl carbon signal of C-3 ketone and appearance of a downfield proton resonance at δ_H_ 3.40 (see Table 1 and Table 2), corresponding to δ_C_ 76.3 signal in the HSQC spectrum, implied a monooxygenation in addition to ketone formation on the cardenolide framework. In the HMBC spectrum, key cross peaks from the carbon at δ_C_ 76.3 to H_2_-11 (δ_H_ 1.30, m; 1.66, m) and H_3_-18 (δ_H_ 0.89, s) helped us to determine the location of hydroxy group as C-12. The relative configuration of C-12 was established via 2D-NOESY data. The strong correlation between H-12 (δ_H_ 3.40, dd, *J* = 11.8, 4.1 Hz) and α- oriented H-17 (δ_H_ 3.59 d, *J* = 7.6 Hz), was evident to establish the orientation of C-12(OH) as *β*. On the basis of these evidence, **7** was identified as 3-oxo-diginatigenin, a known compound [37].

The HR-ESI-MS spectrum of compound **8** provided a major ion peak at *m*/*z* 405.2280 [M-H]^−^ (calcd. 405.2355 for C_23_H_34_O_6_ [M-H]^−^), indicating 16 amu increase over gitoxigenin and **5** (3-epi-gitoxigenin). The ^1^H- and ^13^C-NMR spectra of **8** showed substantial resemblance with those of **5** rather than of gitoxigenin (Table 1 and Table 2). The ^1^H-NMR spectrum of **8**, in addition to the characteristic low-field signals of **5**, an additional resonance at δ_H_ 3.34 was observed. In the ^13^C-NMR spectrum, the additional oxymethine signal at δ_C_ 76.6 undoubtedly demonstrated monohydroxylation on the skeleton, while a key HMBC between the new oxymethine carbon signal and H_3_-18 (δ_H_ 0.87, s) helped us locating the hydroxy group at C-12. In the NOESY spectrum, H-12 (δ_H_ 3.34, m) showed correlation with α-oriented H-16 (δ_H_ 4.60, td, *J* = 7.6, 1.7 Hz), establishing the orientation of OH-12 as *β*. Also, the NOESY correlation between H-3 (δ_H_ 3.59, m) and *β*-oriented H-5 (δ_H_ 1.45, m) verified the epimerization at C-3 indicating α-orientation of C-3(OH). In conclusion, the structure of **8** was elucidated as 3-epi-diginatigenin.

The microbial biotransformation study on gitoxigenin afforded five new cardenolide derivatives **1**, **3**, **4**, **6** and **8** together with 3 previously identified compounds **2**, **5** and **7**. The structures and approximate yields were illustrated in Figure 5.

Structurally, the 3-oxo metabolites of gitoxigenin **1**–**3** and **7** were mainly obtained. The formation of 3-oxogitoxigenin by *Gibberella fujikuroi* and *Fusarium* sp. was demonstrated in earlier biotransformation studies [34,36,37,38]. The structure was determined using paper partition chromatography, IR and Kedde’s color reaction; however, NMR-based structural characterization was absent. Also, absorption spectroscopy analysis of gitoxigenin and 3-oxo-gitoxigenin was reported with a series of related cardenolides. Likewise, other known 3-keto derivative 3-oxodiginatigenin (**7**), was fermentatively prepared from diginatigenin previously by Repke and Klesczewski; however, limited spectroscopic data was available [37]. Thus, in this study, **2** and **7** were characterized thoroughly based on the NMR data for the first time.

Compounds **5**, **6** and **8** were 3-epi derivatives of gitoxigenin. Similar epimerization metabolites were earlier reported in independent biotransformation and semi-synthesis studies [36,37,38,39]. Moreover, Repke and Samuels carried out an in vitro study with liver tissue preparations from rats using digitoxigenin as substrate to examine the enzymatic foundation of C-3 epimerization [40]. It was pointed out that the epimerization process of cardiac steroids involved two stages: first, oxidation of the substrate by diphosphopyridine nucleotide (DPN)-dependent 3*β*-hydroxy steroid dehydrogenases; second, reduction of the 3-keto intermediate by the triphosphopyridine nucleotide (TPN)-dependent 3*α*-hydroxy steroid dehydrogenases. Also, Hennebert et al. carried out docking experiments on steroids with intention to explain the epimerization mechanism of dehydrogenases [41]. Considering the earlier reports and the formation of 3-keto metabolites chiefly in this study, it is rational to state that the 3-epi derivatives are likely to be originated via 3-keto intermediates by 3*α*-hydroxy dehydrogenases.

Finally, compound **4** exhibited an unusual structure with dimethyl acetal functionality at C-3 reported here for the first time. Previously, 3,3-dimethoxy analogs of two cardenolides (uzarigenin and digitoxigenin) were prepared via semi-synthesis [42]. It is known that aldehydes and ketones in acidic environment with the presence of methanol can form dimethyl acetals. To make sure that **4** is not an artefact, LC-MS analyses were performed for the samples of Day 6, 10, 14, 18 and the final ethyl acetate (EtOAc) extract (Day 21) used for isolation studies (see Appendix A). In these experiments, the extracts were treated with neither acids nor methanol. Since compound **4** was detected in the last three time points (Day 14, 18 and 21), the formation of dimethyl acetal was suggested to take place via biocatalysis. One plausible path for acetal formation is P450 monoxygenase catalyzed geminal-diol formation at C-3. Accordingly, Bell-Parikh and Guengerich demonstrated a slow oxidation reaction of ethanol to acetic acid by human P450 2E1 and showed that gem-diol could be formed as an intermediate within the reaction catalyzed via monooxygenation [43]. In another study, gem-diol formation was catalyzed by CYP112, a bacterial P450 monooxygenase, as a part of microbial gibberellin synthesis [44]. Based on the abovementioned research, a plausible stepwise mechanism for acetal formation on the cardenolide core is put forward to be catalyzed by P450 monooxygenases (Figure 6).

Regarding hydroxylation, the monooxygenation took place only on the *β* face of the steroid nucleus involving C-7 (**1**, **3**, **4** and **6**) and C-12 (**7** and **8**) positions (Figure 6). In other words, stereoselective monooxygenations were afforded by *A. eureka* endophyte on the cardenolide skeleton. Most metabolites were monohydroxylated derivatives, whereas the only dihydroxy product was compound **3**.

Monooxygenations on the positions of C-7 and C-12 are frequently observed on cardenolides [45]. Diginatigenin, also known as 12*β*-hydroxygitoxigenin, was reported first by Murphy, as the aglycone of a cardiac glycoside diginatin isolated from *Digitalis lanata* [46,47], and later Titus et al. isolated the compound as a biotransformation product of gitoxigenin by *Fusarium lini*, reconfirming the structure [28]. Our biotransformation studies on gitoxigenin by *A. eureka* yielded isolation of 3-epi analog of diginatigenin (**8**, 3-epi-diginatigenin), a previously unidentified compound.

Understanding the metabolism of drugs is vital for preclinical and clinical phases of drug discovery processes being one of the most important factors affecting efficacy/safety rate. Cardenolides, for this purpose, were subjects of several microbial transformation studies, in which some oxidized metabolites were reported as tentative mammalian metabolites. Repke et al. performed an investigation on animal metabolites of digitoxin. After intravenous administration of digitoxin into rats, urinary and fecal sampling for 48 h led to the isolation of ten compounds, mainly exerting hydrolysis of the glycosidic bonds and 12*β*-hydroxylations [37]. Another study was performed on humans by Ashley et al. through digitoxin administration *per os*, and 12*β*-hydroxylated metabolites were determined in urine samples [48]. Our findings were consistent with both of these studies representing 12*β*-hydroxy metabolites.

### 2.3. MTT Cell Viability Assay

Cytotoxicity of the metabolites and substrate were evaluated in various cancer cell lines (pancreatic: PANC-1, MIA PaCa-2; prostate: DU 145 and lung: A549) and two healthy cell lines (lung: MRC-5 and kidney: HEK-293) by MTT cell viability assay (Table 3).

Compound **7** (3-oxodiginatigenin) exhibited cytotoxic activity against PANC-1 with an IC_50_ value of 6.24 µM, while 3-epidiginatigenin (**8**) demonstrated higher bioactivity against PANC-1, A549, and MIA PaCa-2 with 1.95, 8.25 and 3.4 µM IC_50_ values, respectively. These findings suggested that 12*β*-monohydroxylated derivatives **7** and **8** of gitoxigenin had higher cytotoxicity especially on PANC-1 cell line compared to the other metabolites.

Although, the IC_50_ values of **8** for cancer cell lines were higher than the substrate and the control groups (oleandrin and doxorubicin), slighter cytotoxic effect was observed for healthy cell lines. Therefore, the new compound **8** could be evaluated further because of its better therapeutic index and probably enhanced bioavailability.

Gitoxigenin was recently investigated for its cytotoxic activity against human cancer cell lines; TK-10 (renal adenocarcinoma), UACC-62 (melanoma) and MCF-7 (breast adenocarcinoma), and the IC_50_ values were reported as 0.415 ± 0.03, 2.83 ± 0.17 and 1.80 ± 0.32 µM, respectively [21]. The growth inhibitory effect of gitoxigenin on HeLa cells was also reported as 2.25 µg/mL [49], while in vitro inhibitory activity against human nasopharynx carcinoma was presented with an ED_50_ value of 2.3 µg/mL [50]. According to our results, gitoxigenin exhibited cytotoxic activity against A549, DU 145, PANC-1, MIA PaCa-2 cell lines with IC_50_ values ranging between 1.53 and 4.08 µM.

From the structure activity relationship perspective, as the C-3 epimers of gitoxigenin have reduced cytotoxicity, one can state that the configuration of C-3 (*β*-oriented OH group), as in gitoxigenin, seems important for higher bioactivity. Furthermore, less toxic biotransformation products obtained within this study point out to possible intermediates of detoxification mechanism for gitoxigenin.

Consequently, biotransformation attracts interest because of its advantages, viz. higher specificity and efficiency. This study shows that *A. eureka* 1E1BL1 is a good choice for the biotransformation of cardenolides. As incubation with a specific microorganism can replace a series of reaction steps in chemical synthesis, findings presented here may offer a biotechnological alternative to perform regio- and stereo-selective reactions. Furthermore, the functional enzymes responsible for C-12*β* and C-7*β* stereoselective hydroxylations, C-3 epimerization, C-3 oxidation or dimethyl acetal formation reactions on cardenolides, might be isolated from *A. eureka* and then be used as biocatalysts in metabolite engineering processes with the purpose of decreasing the cost of downstream processing.

## 3. Materials and Methods

### 3.1. General Experimental Procedures

Column chromatography studies were carried out using silica gel 60 (70-230 mesh, Merck, Darmstadt, Germany) and LiChroprep^®^ RP-C18 (25–40 µm particle size, Merck) as adsorbents. A medium-pressure pump was used for vacuum liquid chromatography (VLC) while RP-C_18_ column chromatography was accompanied by a peristaltic pump. Analytical thin layer chromatography (TLC) was carried out on pre-coated silica gel 60 F_254_ or RP-C_18_ F_254_ aluminium plates (20 × 20 cm, layer thickness 0.2 mm, Merck). All isolation and purification steps have been checked by TLC plates on which spots were visualised by spraying with 20% sulphuric acid in water and then heated at 105 °C until visualized under day light and examined under UV light (254 and 365 nm). Mass spectrometry analysis were carried out on an Agilent 1200/6530 HR-ESI-MS instrument (Agilent Technologies, Inc., Santa Clara, CA USA) in positive and negative modes. The LC-MS analysis was performed on an Agilent 1260 Series HPLC system coupled with an Agilent G6550A iFunnel Quadrupole Time-of-Flight with a Dual Spray Agilent Jet Stream electrospray ionisation source (Agilent Technologies, Inc.). The instrument was operated in 2 GHz Extended Dynamic Range mode with negative electrospray ionisation. The separation was performed on a reversed phase Agilent Zorbax Extend column (4.6 mm × 150 mm × 3.5 µm). The data was processed by Agilent MassHunter software B 06.00. 1D-, 2D- NMR spectra were recorded on a Varian AS 400 Mercury NMR spectrometer (Varian, Palo Alto, CA, USA) and a Bruker DRX 500 (Bruker Biospin, Silberstreifen 4, Rheinstetten, Germany). ^1^H-NMR spectra were recorded at 400/500 MHz, and the chemical shifts were reported in units of parts per million (ppm) relative to the solvent peaks (CDCl_3_: δ = 7.26, CD_3_OD: δ = 3.33 ppm or pyridine-*d_5_*: δ = 8.74; 7.58 and 7.22 ppm). ^13^C-NMR spectra were recorded at 100 MHz, and the chemical shifts were reported in ppm relative to the solvent peaks (CDCl_3_: δ = 77.0, CD_3_OD: 49.0 ppm or pyridine-*d_5_*: 150.4; 135.9 and 123.9 ppm). FT-IR (Fourier transform infrared) spectroscopic data were collected on a Perkin-Elmer Spectrum 100 FT-IR Spectrometer (Perkin Elmer, Wellesley, MA, USA). Optical rotations were measured in methanol using a AutoPol I polarimeter (Rudolph Research Analytical, 55 Newburgh Road Hackettstown, NJ, USA) at 18 °C.

### 3.2. Plant Material

*Nerium oleander* L. was harvested from Urla (Izmir, Turkey) in Nov 2019 (38°19′09.4″ N 26°38′35.5″ E). The plant material was identified by Prof. Dr. Erdal BEDİR (Department of Bioengineering, IZTECH, Izmir, Turkey) and a voucher specimen is deposited at the Herbarium of the Department of Pharmacognosy, Faculty of Pharmacy, Ege University, Izmir, Turkey with the herbarium number of 1640.

### 3.3. Preparation of Gitoxigenin, the Substrate, by Acid-Catalysed Hydrolysis of Oleandrin

In the context of this study, oleandrin, which was recently isolated from dried leaves of *N. oleander* according to a modified method of Ryer et al. [51] by our research team [33], has been used as the main starting compound.

An acid-catalysed hydrolysis reaction of oleandrin was performed in order to obtain gitoxigenin. Briefly, 4 g of oleandrin was dissolved in 1500 mL of 2 N trifluoroacetic acid (TFA) in *n*-butanol and stirred at 200 °C for 90 min under reflux. Hydrolysis was carried out in four portions (1g oleandrin in 375 mL 2 N TFA in *n*-butanol per each). The progression of the reaction was checked by TLC (Mobile system: 97:3 chloroform: methanol) at every 30 min. At the end of 90 min, the reaction was ended neutralizing with 4 N KOH in water. Neutral reaction mixture was partitioned between chloroform and water in order to remove salt and other inorganic impurities. Chloroform was evaporated at 40 °C and the organic residue (12.2 g) was loaded onto a silica gel column using a gradient system of chloroform: isopropanol (100:0 to 95:5) to afford 80 fractions. Gitoxigenin with slight impurities was obtained from the collected fractions of 51–72. Further purification was achieved by precipitation of Fr: 51–72 in acetonitrile (ACN) resulting in high purity gitoxigenin (1.05 g, ca. 25% yield) as a white powder. Structural elucidation of gitoxigenin was carried out by modern spectroscopic techniques (1D, 2D NMR and HR-ESI-MS).

### 3.4. Microbial Biotransformation Procedure

Biotransformation of gitoxigenin was performed by *Astragalus angustifolius*-originated endophytic fungus *Alternaria eureka* 1E1BL1, which was isolated and identified by our group in 2016 [52]. The fungal strain stocks prepared as spore solutions in glycerol were stored at −80 °C, also potato dextrose agar (PDA) slants prepared and stored at 4 °C until use. Fungi was grown on potato dextrose agar (PDA) Petri dishes at 25 °C for 10 days, then spore solution was prepared in 0.1% (*v*/*v*) Tween 80 and used to inoculate (2% *v*/*v*) five of 5 L Erlenmeyer flasks containing 1.5 L of potato dextrose broth (PDB). After incubation for 3 days, 1050 mg of gitoxigenin was added into the flasks, and biotransformation was carried out in a rotary shaker at 180 rpm, 25 °C for 21 days. Biotransformation process was followed by checking the ethyl acetate extracts of daily collected samples on TLC plates [29].

### 3.5. Isolation and Purification

After three weeks of biotransformation, fungal biomass was removed by filtering through Whatman No.1 filter paper on Buchner funnel under vacuum, and the broth was extracted three times with equal volume of EtOAc. The organic phase was evaporated by rotary vacuum evaporator at 40 °C.

Initially, the crude EtOAc extract (8.7 g) was subjected to a silica gel column (175 g) and eluted with a gradient system of CHCl_3_: MeOH (98:2, 97:3, 95:5, 90:10). Following collection of the main fractions (1–620), similar ones were combined together based on their TLC profiles.

Firstly, the combined fractions of 15–16 (172 mg) were loaded onto an RP-C_18_ (17.5 g) column which was eluted in gradient mode with H_2_O: ACN (70:30 to 60:40) using a VLC system. Subfractions 14–18 were pooled together and precipitated in cold ACN to obtain **1** (11.5 mg). The main fraction 14 (36.6 mg) was applied to a silica gel column (12 g) and eluted with isocratic CHCl_3_ (100%) yielding **2** (4.6 mg).

Compound **3** (2.6 mg) was isolated from the combined fractions of 528-572 (22.9 mg), chromatographed over RP-C_18_ (17.5 g) by a VLC system using H_2_O: ACN (80:20) for elution. Compound **4** (4.3 mg) was recovered from the combined fractions of 221–280 (71.3 mg) that was loaded onto a silica gel column (40 g) and eluted with a gradient system of CHCl_3_:MeOH (98:2, 97:3, 96:4, 90:10).

The combined fractions of 281–370 (40.4 mg) were applied to a silica gel column (15.7 g) and eluted with an increasing gradient of *n*-hexane: EtOAc: MeOH (10:10:0.5 to 10:10:1.5). Subfractions 54–70 (14.6 mg) was subjected to a sequential fractionation using silica gel (7 g) as stationary phase and CHCl_3_: MeOH (97:3) as mobile phase to obtain **5** (2.3 mg).

Compounds **6** and **7** were isolated from the main fractions of 393–430 (37.4 mg), which was chromatographed over an RP-C_18_ (17.5 g) VLC column and eluted with a gradient system of H_2_O:ACN (from 80:20 to 20:80). Subfraction 24 was further purified through crystallization in cold ACN yielding **6** (1.9 mg), while **7** (2.7 mg) was obtained directly from subfraction 22 in pure form. Compound **8** (1.6 mg) was isolated from the main fractions of 604-619 (5.2 mg), which was fractionated on RP-C_18_ (17.5 g) by VLC system using a gradient system of H_2_O:ACN (from 90:10 to 80:20).

### 3.6. Compound Characterization

*4-((3S*,*5R*,*10S*,*13R*,*14S*,*16S*,*17R)-3*,*14*,*16-Trihydroxy-10*,*13-dimethylhexadecahydro- 1H-cyclopenta[a]phenanthren-17-yl) furan-2(5H)-one*) (gitoxigenin). White powder. FT-IR (CHCl_3_: MeOH): 3526, 3431, 2942, 2887, 1789, 1759, 1737, 1629, 1451, 1376, 1185, 1093, 1033, 888 cm^−1^. HR-ESI-MS (negative ion mode): *m*/*z* = 435.2387 [M + FA-H]^−^ [Calculated for C_23_H_34_O_5_ [M + FA-H]^−^ 435.2388]. [α]_D_^18^= +14.7° (c 0.27, MeOH). ^1^H-NMR (CD_3_OD/drops of pyridine-*d_5_*, 500 MHz): see Table 1; ^13^C-NMR (CD_3_OD /drops of pyridine-*d_5_*, 100 MHz): see Table 2.

(*4-((5R*,*10S*,*13R*,*14S*,*16S*,*17R)-7*,*14*,*16-Trihydroxy-10*,*13-dimethyl-3-oxohexadecahydro-1H-cyclopenta[a]phenanthren-17-yl)furan-2(5H)-one* (7*β*-hydroxy-3-oxo-gitoxigenin, **1**). White needles. FT-IR (CHCl_3_): 3405, 2942, 2873, 2156, 1739, 1712, 1631, 1617, 1449, 1341, 1284, 1110, 1051, 1032, 755 cm^−1^. HR-ESI-MS (negative ion mode): *m*/*z* = 403.2123 [M − H]^−^ [Calculated for C_23_H_32_O_6_ [M-H]^−^ 403.2126]. [α]_D_^18^= +20.4° (c 0.25, MeOH). ^1^H-NMR (CDCl_3_/1 drop of CD_3_OD, 400 MHz): see Table 1. ^13^C-NMR (CDCl_3_/1 drop of CD_3_OD, 100 MHz): see Table 2.

(*4-((5R*,*10S*,*13R*,*14S*,*16S*,*17R)-14*,*16-Dihydroxy-10*,*13-dimethyl-3-oxohexadecahydro-1H-cyclopenta[a]phenanthren-17-yl)furan-2(5H)-one* (3-oxogitoxigenin, **2**): White crystalline powder. FT-IR (CHCl_3_: MeOH): 3443, 2926, 2863, 1739, 1714, 1615, 1447, 1285, 1109, 1032, 895 cm^−1^. HR-ESI-MS (negative ion mode): *m*/*z* = 433.2227 [M + FA-H]^−^ [Calculated for C_23_H_32_O_5_ [M + FA-H]^−^ 433.2231]. [α]_D_^18^ = +12.5° (c 0.4, MeOH). ^1^H-NMR (CDCl_3_, 400 MHz): see Table 1. ^13^C-NMR (CDCl_3_, 100 MHz): see Table 2.

*(4-((5R,10S,13S,14S,16S,17R)-7,12,14,16-Tetrahydroxy-10,13-dimethyl-3-oxohexadecahydro-1H-cyclopenta[a]phenanthren-17-yl)furan-2(5H)-one* (7*β*,12*β*-dihdyroxy-3-oxogitoxigenin, **3**): Yellowish-white oily substance. FT-IR (CHCl_3_: MeOH): 3381, 2935, 2876, 1732, 1628, 1448, 1277, 1110, 1051, 1014 cm^−1^. HR-ESI-MS (negative ion mode): *m*/*z* = 419.2075 [M − H]^−^ [Calculated for C_23_H_32_O_7_ [M − H]^−^ 419.2075]. [α]_D_^18^= +12.7° (c 0.24, MeOH). ^1^H-NMR (CD_3_OD, 400 MHz): see Table 1. ^13^C-NMR (CD_3_OD, 100 MHz): see Table 2.

*(4-((5R,10S,13R,14S,16S,17R)-7,14,16-Trihydroxy-3,3-dimethoxy-10,13-dimethylhexadecahydro-1H-cyclopenta[a]phenanthren-17-yl)furan-2(5H)-one* (3-dimethylacetal-7*β*-hydroxygitoxigenin, **4**): White powder. FT-IR (CHCl_3_): 3382, 2937, 2874, 1736, 1616, 1450, 1363, 1279, 1178, 1107, 1094, 1051, 1035, 877, 755 cm^−1^. HR-ESI-MS (negative ion mode): *m*/*z* = 449.2541 [M + FA-H]^−^ [Calculated for C_25_H_38_O_7_ [M + FA-H]^−^ 449.2545]. [α]_D_^18^ = +20° (c 0.2, MeOH). ^1^H- NMR (CDCl_3_, 400 MHz): see Table 1. ^13^C-NMR (CDCl_3_, 100 MHz): see Table 2.

(*4-((3R*,*5R*,*10S*,*13R*,*14S*,*16S*,*17R)-3*,*14*,*16-Trihydroxy-10*,*13-dimethylhexadecahydro-1H-cyclopenta[a]phenanthren-17-yl)furan-2(5H)-one* (3-epigitoxigenin, **5**): White needles. FT-IR (CHCl_3_: MeOH): 3411, 2935, 2865, 2156, 1735, 1615, 1450, 1379, 1170, 1114, 1093, 1073, 1034 cm^−1^. HR-ESI-MS (negative ion mode): *m*/*z* = 435.2385 [M + FA-H]^−^ [Calculated for C_23_H_34_O_5_ [M + FA-H]^−^ 435.2388]. [α]_D_^18^= + 9.2° (c 0.11, MeOH). ^1^H-NMR (CD_3_OD, 400 MHz): see Table 1.; ^13^C-NMR (CD_3_OD, 100 MHz): see Table 2.

(*4-((3R*,*5S*,*10S*,*13R*,*14S*,*16S*,*17R)-3*,*7*,*14*,*16-Tetrahydroxy-10*,*13-dimethylhexadecahydro-1H-cyclopenta[a]phenanthren-17-yl)furan-2(5H)-one* (7*β*-hydroxy-3-epigitoxigenin, **6**): White needles. FT-IR (CHCl_3_: MeOH): 3391, 2936, 2869, 2156, 1734, 1455, 1177, 1112, 1051, 774 cm^−1^. HR-ESI-MS (negative ion mode): *m*/*z* = 405.2280 [M − H]^−^ [Calculated for C_23_H_34_O_6_ [M − H]^−^ 405.2283]. [α]_D_^18^ = +6.9° (c 0.29, MeOH). ^1^H-NMR (pyridine-d_5_, 400 MHz): see Table 1. ^13^C-NMR (pyridine-*d_5_*, 100 MHz): see Table 2.

(*4-((5R*,*10S*,*13S*,*14S*,*16S*,*17R)-12*,*14*,*16-Trihydroxy-10*,*13-dimethyl-3-oxohexadecahydro-1H-cyclopenta[a]phenanthren-17-yl)furan-2(5H)-one* (3-oxodiginatigenin, **7**): White powder. FT-IR (CHCl_3_: MeOH): 3394, 2924, 2865, 1733, 1713, 1630, 1455, 1287, 1019, 775 cm^−1^. HR-ESI-MS (negative ion mode): *m*/*z* = 449.2179 [M + FA-H]^−^ [Calculated for C_23_H_32_O_6_ [M + FA-H]^−^ 449.2181]. [α]_D_^18^= +21.9° (c 0.09, MeOH). ^1^H-NMR (CD_3_OD, 400 MHz): see Table 1. ^13^C- NMR (CD_3_OD, 100 MHz): see Table 2.

(*4-((3R*,*5R*,*10S*,*13S*,*14S*,*16S*,*17R)-3*,*12*,*14*,*16-Tetrahydroxy-10*,*13-dimethylhexadecahydro-1H-cyclopenta[a]phenanthren-17-yl)furan-2(5H)-one* (3-epidiginatigenin, **8**): White powder. FT-IR (CHCl_3_: MeOH): 3390, 2929, 2863, 1732, 1614, 1559, 1451, 1074, 1035, 1027 cm^−1^. HR-ESI-MS (negative ion mode): *m*/*z* = 405.2280 [M − H]- [Calculated for C_23_H_34_O_6_ [M − H]^−^ [405.2355]. [α]_D_^18^ = 0.0° (c 0.18, MeOH). ^1^H-NMR (CD_3_OD, 400 MHz): see Table 1. ^13^C-NMR (CD_3_OD, 100 MHz): see Table 2.

### 3.7. MTT Assay

To see cytotoxic effects of the obtained metabolites, a standard MTT (3-(4,5-dimethylthiazol-2-yl)-2,5-diphenyltetrazolium bromide) cell viability assay was performed on human cancer cell lines PANC-1 (pancreatic, ATCC^®^ CRL-1469™), MIA PaCa-2 (pancreatic, ATCC^®^ CRL-1420™), DU 145 (prostate, ATCC^®^ HTB-81™) and A549 (lung, ATCC^®^ CCL-185™) and two healthy cell lines MRC-5 (lung, ATCC^®^ CCL-171™) and HEK-293 (kidney, ATCC^®^ CRL-1573™) according to manufacturer’s instructions (M2128, Sigma-Aldrich^®^). For this purpose, cells at 80–90% confluency were seeded to 96-well plates as 5-10x10^3^ cells/well in Eagle Minimal Essential Medium (EMEM) or Dulbecco Modified Eagle Medium (DMEM); supplemented with 10% FBS and 1% L-glutamine depending on the cell type and incubated in a humidified incubator at 37 °C with 5% CO_2_ atmosphere for 24 h. After cell adhesion, gitoxigenin and the metabolites were applied to the cells in triplicate with concentrations ranging between 0.001 and 10 µM. After 48 h exposure to certain concentrations of the compounds, medium was replaced with fresh one containing 10% MTT and plates were incubated for 3 h in the dark for the conversion of tetrazolium salts to formazan crystals by mitochondrial metabolism of viable cells. After incubation, formazan crystals were dissolved in DMSO and absorbance of each well were measured at 570 nm via microplate reader (Synergy™ HTX- BioTek, Winooski, VT, USA). Doxorubicin (10 nM, 1 µM and 4 µM) was used as positive control, whereas oleandrin (1 nM, 10 nM and 100 nM) was the second positive control as a potent cardenolide.

All compounds were dissolved in DMSO (Sigma Aldrich, St. Louis, MO, USA) and the final DMSO concentration applied to cells was 0.02% (*v*/*v*), which is non-toxic to cells. The IC_50_ values of the compounds were calculated by the normalization of optical densities (OD) to negative control DMSO (0.02%, *v*/*v*).

## Figures and Tables

**Figure 1 molecules-26-03030-f001:**
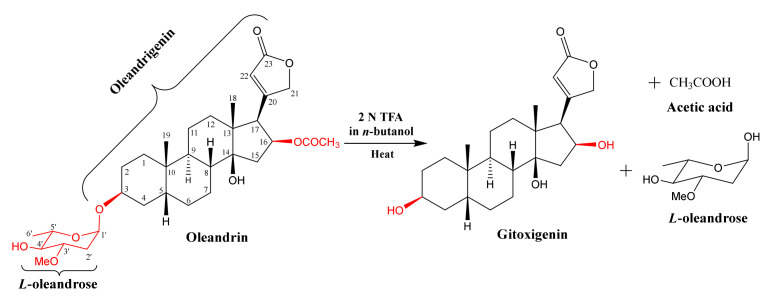
Acid-catalysed hydrolysis of cardiac glycoside oleandrin and structures of oleandrin, gitoxigenin and *L*-oleandrose (TFA: trifluoroacetic acid).

**Figure 2 molecules-26-03030-f002:**
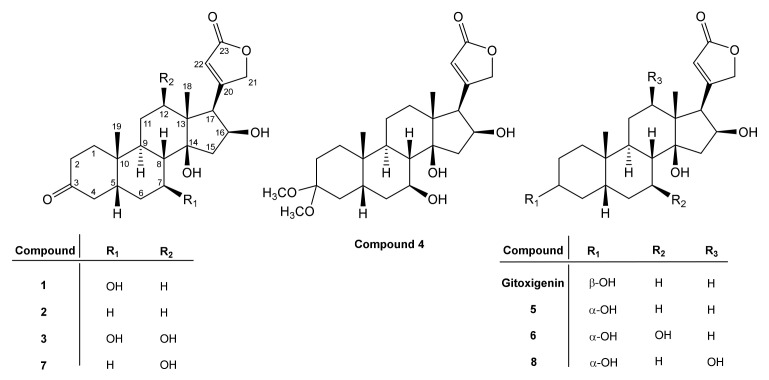
Chemical structures of gitoxigenin (substrate) and eight biotransformation products transformed by *A. eureka* 1E1BL1.

**Figure 3 molecules-26-03030-f003:**
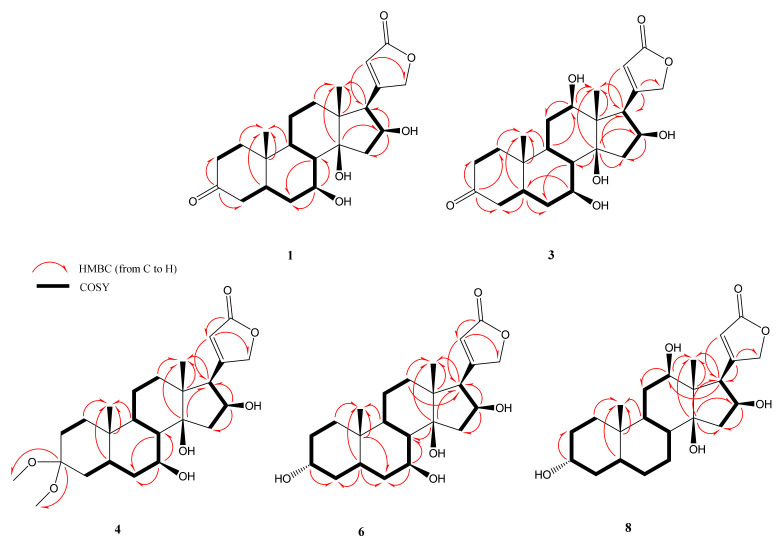
Key HMBC and COSY correlations of the new metabolites (compounds **1**, **3**, **4**, **6** and **8**) obtained from the microbial biotransformation of gitoxigenin by *A. eureka* 1E1BL1.

**Figure 4 molecules-26-03030-f004:**
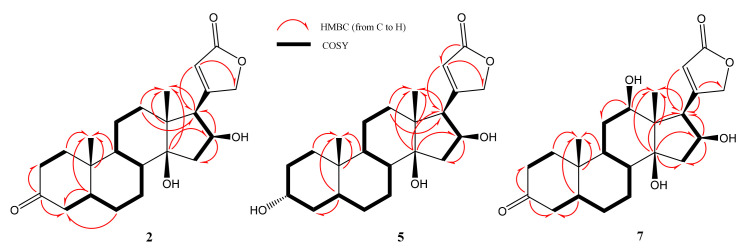
Key HMBC and COSY correlations of the known metabolites (compounds **2**, **5** and **7**) obtained from the microbial biotransformation of gitoxigenin by *A. eureka* 1E1BL1.

**Figure 5 molecules-26-03030-f005:**
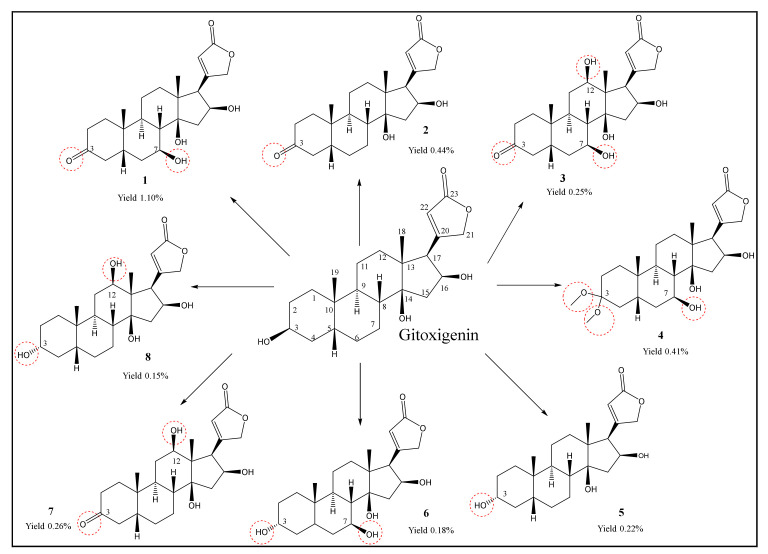
Structures and yields of the metabolites obtained via microbial biotransformation of gitoxigenin by *A. eureka* 1E1BL1.

**Figure 6 molecules-26-03030-f006:**
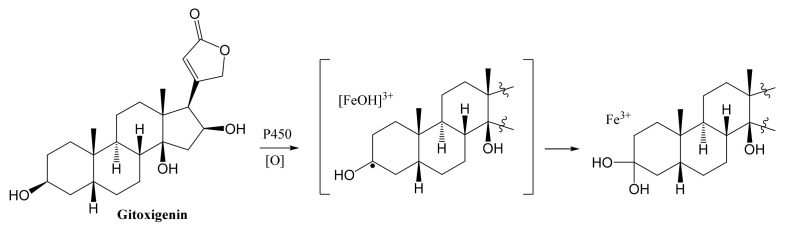
A plausible path for the formation of acetal group in **4**.

**Table 1 molecules-26-03030-t001:** ^1^H NMR data of gitoxigenin and compounds **1–8** (δ_H_ (ppm), (*J*, Hz)).

Position	Gitoxigenin	1	2	3	4	5	6	7	8
**1**	1.22 m; 1.27 m	1.41 m; 1.93 m	1.45 m; 2.00 m	1.48 m; 2.05 dd (14.2,4.5)	1.25 m; 1.65 m	1.04 td (14.2,3.5); 1.82 m	1.06 d (14.5); 1.80 m	1.45 m; 2.11 m	1.11 td (14.2, 3.4); 1.85 m
**2**	1.24 m; 1.36 m	2.12 m; 2.21 td (14.4, 5.3)	2.20 d (16.1); 2.32 dd (14.5, 6.4)	2.12 m; 2.47 td (14.5, 5.1)	1.26 m; 1.77 m	1.37 m; 1.65 m	1.65 q (12.5); 1.96 m	2.12 m; 2.57 dd (14.4, 5.5)	1.39 m; 1.69 m
**3**	3.82 t (2.8)	-	-	-	-	3.56 m	3.85 m	-	3.59 m
**4**	1.09 m; 1.67 td (13.7, 3.2)	2.10 m; 2.43 m	2.06 m; 2.62 m	2.12 m; 2.66 t (14.3)	1.55 m; 1.77 m	1.49 m; 1.74 m	1.92 m; 1.99 m	1.97 m; 2.81 t (14.3)	1.53 m; 1.70 m
**5**	1.54 m	1.86 m	1.84 m	1.88 m	1.67 m	1.40 m	1.51 m	1.82 m	1.45 m
**6**	0.96 m; 1.63 m	1.50 m; 1.83 m	1.37 m; 1.90 m	1.58 m; 1.90 m	1.57 m; 1.92 m	1.35 m; 1.88 m	1.80 m; 2.14 td (12.2, 4.9)	1.37 m; 1.91 m	1.38 m; 1.91 m
**7**	0.97 m; 1.57 m	3.84 td (10.6, 4.9)	1.24 m; 1.91 m	3.91 td (11.6, 11.2, 4.7)	3.96 m	1.29 m; 1.84 m *	4.26 m	1.31 m; 1.91 m	1.31 m; 1.84 m
**8**	1.34 m	1.69 dd (11.8, 10.4)	1.62 m	1.71 m	1.72 m	1.59 m	1.95 m	1.65 m	1.60 m
**9**	1.37 m	1.58 m	1.69 m	1.88 m	1.54 m	1.70 m	1.77 m	1.89 m	1.72 m
**10**	-	-	-	-	-	-	-	-	-
**11**	0.94 m; 1.11 m	1.31 m; 1.37 m	1.32 m; 1.46 m	1.45 m; 1.63 m	1.25 m; 1.41 m	1.21 m; 1.42 m *	1.25 m; 1.35 m	1.30 m; 1.66 m	1.30 m; 1.61 m
**12**	1.11 m; 1.26 m	1.25 m; 1.61 m	1.35 m; 1.69 m	3.36 dd (11.8, 4.1)	1.25 m; 1.65 m	1.39 m; 1.55 m	1.31 m; 1.49 m	3.40 dd (11.8, 4.1)	3.34 m
**13**	-	-	-	-	-	-	-	-	-
**14**	-	-	-	-	-	-	-	-	-
**15**	1.49 dd (15.0, 2.4); 2.35 dd (14.9, 8.5)	1.95 m; 2.36 dd (14.2, 6.4)	1.90 m; 2.42 dd (14.5, 6.4)	1.97 m; 2.59 dd (14.5, 7.5)	2.05 m; 2.40 dt (14.4, 4.8)	1.72 m; 2.63 dd (14.9, 8.5)	2.27 d (13.8); 2.67 m	1.79 m; 2.49 dd (14.9, 7.6)	1.82 m; 2.43 dd (14.7, 7.6)
**16**	4.43 td (8.3, 2.4)	4.41 t (6.5)	4.54 t (6.5)	4.58 m	4.44 brs	4.66 td (8.2, 2.3)	4.94 m	4.59 s	4.60 td (7.6, 1.7)
**17**	2.88 d (7.9)	2.91 d (6.9)	2.96 d (7.1)	3.58 d (7.6)	2.91 dd (7.0, 3.0)	3.14 d (7.9)	3.29 d (8.0)	3.59 d (7.6)	3.59 m
**18**	0.66 s	0.90 s	0.98 s	0.89 s	0.96 s	0.92 s	1.11 s	0.89 s	0.87 s
**19**	0.66 s	0.98 s	1.02 s	1.06 s	0.97 s	0.93 s	0.93 s	1.06 s	0.97 s
**20**	-	-	-	-	-	-	-	-	-
**21**	4.96 d (18.4); 4.91 m	4.87 d (18.3); 5.02 d (18.3)	4.90 d (18.1); 5.05 d (18.1)	5.03 dd (18.5, 1.8); 5.14 dd (18.5, 1.9)	4.89 m; 5.06 m	5.11 dd (18.4, 1.8); 5.17 dd (18.5, 1.8)	5.57 m; 5.69 m	5.04 dd (18.3, 1.8); 5.18 dd (18.4, 1.9)	5.04 dd (18.5, 1.8); 5.19 dd (18.5, 1.9)
**22**	5.71 s	5.88 s	5.97 s	6.01 s	5.97 s	5.94 s	6.25 s	6.03 s	6.03 s
**23**	-	-	-	-	-	-	-	-	-
**3-*O*-Me**	-	-	-	-	3.12 s	-	-	-	-
**3-*O*-Me**	-	-	-	-	3.19 s	-	-	-	-
**Solvent**	CD_3_OD/a drop of pyridine-*d_5_*	CDCl_3_/a drop of CD_3_OD	CDCl_3_	CD_3_OD	CDCl_3_	CD_3_OD	Pyridine-*d_5_*	CD_3_OD	CD_3_OD

* interchangeable.

**Table 2 molecules-26-03030-t002:** ^13^C-NMR data of gitoxigenin and compounds **1**–**8** δ_C_ (ppm).

Position	Gitoxigenin	1	2	3	4	5	6	7	8
**1**	30.7 t	36.0 t	36.7 t	37.6 t	32.4 t	36.2 t	35.7 t	37.8 t	36.2 t
**2**	28.5 t	36.8 t	37.2 t	38.9 t	27.5 t	31.3 t	31.8 t	37.9 t	31.2 t
**3**	67.6 d	212.7 s	212.7 s	214.5 s	100.4 s	72.3 d	71.2 d	215.9 s	72.2 d
**4**	34.1 t	42.8 t	42.2 t	43.7 t	34.5 t	37.0 t	38.7 t	43.0 t	37.0 t
**5**	37.3 d	43.7 d	43.6 d	45.1 d	40.1 d	43.1 d	43.1 d	45.4 d	43.1 d
**6**	27.7 t	35.9 t	26.6 t	37.2 t	37.1 t	28.2 t	38.5 t	27.7 t	28.2 t
**7**	22.4 t	69.1 d	21.5 t	70.3 d	70.9 d	22.6 t *	70.5 d	22.4 t	23.0 t
**8**	42.8 d	45.6 d	41.7 d	46.8 d	46.4 d	43.0 d	47.3 d	42.1 d	42.4 d
**9**	36.5 d	35.8 d	36.7 d	33.8 d	35.0 d	37.4 d	36.2 d	34.4 d	34.3 d
**10**	36.3 s	35.0 s	35.4 s	36.0 s	35.1 s	35.9 s	35.4 s	36.2 s	35.8 s
**11**	22.0 t	21.3 t	21.2 t	30.7 t	21.4 t	21.9 t *	21.5 t	30.7 t	30.5 t
**12**	40.8 t	41.2 t	41.5 t	75.9 d	42.0 t	40.9 t	40.2 t	76.3 d	76.6 d
**13**	51.3 s	49.5 s	49.8 s	57.2 s	49.4 s	51.3 s	50.5 s	57.6 s	57.6 s
**14**	85.5 s	86.0 s	85.8 s	87.0 s	86.6 s	85.6 s	85.4 s	86.3 s	86.4 s
**15**	43.8 t	42.1 t	42.2 t	44.4 t	42.4 t	43.8 t	45.5 t	43.4 t	43.5 t
**16**	73.0 d	73.0 d	73.2 d	73.6 d	73.7 d	73.1 d	72.7 d	73.4 d	73.4 d
**17**	59.6 d	58.2 d	58.1 d	55.0 d	58.5 d	59.7 d	59.9 d	54.9 d	54.9 d
**18**	17.1 q	16.8 q	16.9 q	10.5 q	17.0 q	17.0 q	17.4 q	10.5 q	10.5 q
**19**	24.3 q	22.4 q	22.7 q	22.7 q	23.2 q	23.8 q	23.8 q	22.8 q	23.7 q
**20**	173.8 s	169.9 s	168.8 s	173.1 s	169.0 s	173.7 s	173.0 s	173.2 s	173.3 s
**21**	77.9 t	75.9 t	75.7 t	77.7 t	75.6 t	77.9 t	77.2 t	77.7 t	77.7 t
**22**	120.6 d	119.5 d	120.0 d	120.7 d	119.7 d	120.6 d	120.5 d	120.6 d	120.6 d
**23**	177.3 s	175.3 s	174.5 s	177.5 s	174.7 s	177.4 s	175.1 s	177.6 s	177.4 s
**3-*O*-Me**	-	-	-	-	47.7 q (×2)	-	-	-	-

* interchangeable.

**Table 3 molecules-26-03030-t003:** Cytotoxic activities of compounds **1**–**8** and the substrate gitoxigenin.

Compound	A549	DU 145	PANC-1	MIA PaCa-2	HEK-293	MRC-5
**1**	>10	>10	>10	>10	>10	>10
**2**	>10	>10	>10	>10	>10	>10
**3**	>10	>10	>10	>10	>10	>10
**4**	>10	>10	>10	>10	>10	>10
**5**	>10	>10	>10	>10	>10	>10
**6**	>10	>10	>10	>10	>10	>10
**7**	>10	>10	6.24	>10	>10	>10
**8**	8.25	>10	1.95	3.4	>10	9.35
**Gitoxigenin**	2.19	4.08	1.53	3.1	7.6	0.815
**Oleandrin**	0.0426	0.0210	0.0387	0.0350	0.0730	0.0173
**Doxorubicin**	0.6200	0.1900	>10	0.5900	0.5870	0.8650

IC_50_ values are the concentration (µM) required 50% cell viability inhibition for a given compound with a 48 h treatment and were calculated with using cell viability (%) formula via nonlinear regression analysis. Experiment was carried out in triplicate (*n* = 3). IC_50_ values were given for A549 human lung cancer cell line, DU 145 human prostate cancer cell line, PANC-1 human pancreatic cancer cell line, Mia PaCa-2 pancreatic cancer cell line. HEK-293 human embryonic kidney cell line and MRC-5 human lung fibroblast cell line as healthy cell lines. Doxorubicin and oleandrin were used as positive controls.

## Data Availability

The data presented in this study are available in the text and Appendix A. Raw data will be provided by the corresponding author if requested.

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
