# Peer review of "New Cardenolides from Biotransformation of Gitoxigenin by the Endophytic Fungus Alternaria eureka 1E1BL1: Characterization and Cytotoxic Activities"

_molecules, 2021, doi:10.3390/molecules26103030_

Round 1
Reviewer 1 Report
The article presents the biotransformation of a cardenolide by a microorganism. Several metabolites are obtained but several weaknesses can be identified and the general interest of the manuscript needs to be improved. Indeed, the biological activities of metabolites are weak, so the bioconversion aspect needs to be improved.
- The results presented correspond to a single experiment. It was made over a very long time without justification; the quantities and yields of the products obtained are very low, the amount of substrate recovered should be given in order to determine the rate of biotransformation. Optimization experiments are needed.
- Biotransformation was followed by CCM analysis. Yet HPLC chromatograms are shown in supplementary part for crude extract and for some purified products, therefore precise kinetic analysis is possible. It would allow:
- to highlight the pathways for the formation of different products
- perhaps to determine incubation times to obtain intermediates in sufficient quantities to use them as substrates
- I am not convinced that product 4 is a biotransformation metabolite and the proposed pathway for its formation is questionable.
- Contrary to the authors' claims, the chromatograms in Figure S1 do not show the presence of product 4 in the crude extract. Indeed, if we compare the peaks corresponding to products 4, 5 and 6 in the crude extract, the intensity of the peak at t = 10.7 min in the chromatogram S1a (corresponding to product 4) is far too low than is expected based on the quantities of products obtained after purification. Product 4 is obtained after purification with a mass of 4.3 mg and products 5 and 6 with masses of 2.3 mg and 1.9 mg respectively. There are twice as many 4 as other products. The difference in structure cannot explain such a difference in absorption at 210nm.
An LC/MS study should be done to ensure that product 4 is present in the incubation medium before any treatment.
- Regarding its formation, the gem-diol form proposed in Figure 6 is the hydrated form of the ketone, and its formation can be done spontaneously and its dimethylation by SAM remains very hypothetical. Contrary to what the authors indicate, in reference 27 there is no proposed mechanism for an O-methylation by SAM.
- The experimental part corresponding to the isolation and purification of the products is complex and a lot of chromatographies have been necessary. This represents a lot of work but the description is not clear enough in the sense that the elution order of products was not respected in the presentation. A pattern of purification in supplementary part is necessary.
- NMR data are very important, but comparisons are impossible on the one hand because some spectra are lacking in Tables 1 and 2, particularly substrate and product 5, and secondly because all NMR spectra were not made in the same solvent, especially for products 6 and 8 which are isomers in 3-position of substrate and not spectra of compound 5.
Author Response
|
First of all, we would like to thank to the reviewer, who raised very constructive comments to increase the quality of the manuscript. Please see his/her critiques and our answers below. Critique 1. The results presented correspond to a single experiment. It was made over a very long time without justification; the quantities and yields of the products obtained are very low, the amount of substrate recovered should be given in order to determine the rate of biotransformation. Optimization experiments are needed. Reply 1. The duration of this single experiment was established based on our previous studies with steroidal compounds and A. eureka, and the main purpose was to obtain as many metabolites as possible. At this point we are not able to carry out (or design) any optimization experiment. However, an LC-MS analysis was carried out. The analysis demonstrated that the substrate was completely metabolized before 14th day of incubation, signifying that the provided yields were correctly defined for the metabolites. Critique 2. Biotransformation was followed by CCM analysis. Yet HPLC chromatograms are shown in supplementary part for crude extract and for some purified products, therefore precise kinetic analysis is possible. It would allow: - to highlight the pathways for the formation of different products - perhaps to determine incubation times to obtain intermediates in sufficient quantities to use them as substrates Reply 2. We thank to the reviewer for both suggestions. Together with the TLC analyses performed during experiments, the LC-MS analysis was evaluated for a tentative pathway (See Chromatograms S2-S12; in which four available metabolites (1, 2, 4 and 5) with the substrate were screened by LC-MS). The first time point of sampling was Day 6. At that point, we observed compounds 1, 2, 3 and 7. Thus it is difficult to deduce the precise pathway and present in the manuscript. For the second critique, we will keep it in mind to use intermediates as substrates, which would be very valuable. Critique 3. I am not convinced that product 4 is a biotransformation metabolite and the proposed pathway for its formation is questionable. - Contrary to the authors' claims, the chromatograms in Figure S1 do not show the presence of product 4 in the crude extract. Indeed, if we compare the peaks corresponding to products 4, 5 and 6 in the crude extract, the intensity of the peak at t = 10.7 min in the chromatogram S1a (corresponding to product 4) is far too low than is expected based on the quantities of products obtained after purification. Product 4 is obtained after purification with a mass of 4.3 mg and products 5 and 6 with masses of 2.3 mg and 1.9 mg respectively. There are twice as many 4 as other products. The difference in structure cannot explain such a difference in absorption at 210nm. An LC/MS study should be done to ensure that product 4 is present in the incubation medium before any treatment. Reply 3. We thank again the reviewer for carefully going through the manuscript. To confirm that compound 4 is not an artifact, an LC-MS analysis was carried out. Please note that during crude extract preparation and/or liquid chromatography runs, no methanol was used. The LC-MS extracted ion chromatograms clearly shows that 4 is a minor metabolite of biotransformation (See Chromatograms S10-11). It forms around Day 14, and its amount slightly increases in continuation of the incubation. For second part of the critique, we unfortunately disagree with the reviewer regarding the quantities. In purification studies, the fractions in the purest forms are combined and the amounts/yields are reported. The side fractions containing the same compound in impure form are not taken into consideration for quantification (or yield calculations). Therefore, it is not rational to reach a conclusion and state that the difference is not understandable. On the other hand, it would be logical to perform quantitative analyses by HPLC after developing efficient methods in the future, and we plan to do so. Critique 4. Regarding its formation, the gem-diol form proposed in Figure 6 is the hydrated form of the ketone, and its formation can be done spontaneously and its dimethylation by SAM remains very hypothetical. Contrary to what the authors indicate, in reference 27 there is no proposed mechanism for an O-methylation by SAM. Reply 4. Thanks for the comment. We agree with the reviewer. Involvement of the SAM is too speculative. Thus, we removed methylation step from Figure 6. Critique 5. The experimental part corresponding to the isolation and purification of the products is complex and a lot of chromatographies have been necessary. This represents a lot of work but the description is not clear enough in the sense that the elution order of products was not respected in the presentation. A pattern of purification in supplementary part is necessary. Reply 5. Thanks for the comment. An isolation scheme was prepared and added to the supplementary part (See Scheme 1 and 2). Critique 6. NMR data are very important, but comparisons are impossible on the one hand because some spectra are lacking in Tables 1 and 2, particularly substrate and product 5, and secondly because all NMR spectra were not made in the same solvent, especially for products 6 and 8 which are isomers in 3-position of substrate and not spectra of compound 5. Reply 6. As suggested, NMR data of the substrate and metabolite 5 were added to Tables 1 and 2. Regarding use of different solvents, the solubility was our major concern. Because of the minor quantities of the metabolites, the NMR solvents affording no precipitates were preferred to obtain better spectra.
|
Reviewer 2 Report
In this article, authors have analytically and numerically demonstrated the formation of new cardenolides through biotransformation of gitoxigenin by endophytic fungus alternaria eureka 1E1BL1 and evaluated their characterization and cytotoxic activities.
- This manuscript is written in an easy-to-follow, pedagogical style. For each
feature observed in the results, all possible explanations are clearly laid
out and eliminated one by one. In cases where the results do not distinguish
between different scenarios, the consequences of each are comparatively
discussed. The work is very well-cited throughout.
- Presented results seem to be of good quality, and are explained carefully and checked for artifacts.
I have only one remark for the abstract where author needs to mention the incubation period with Endophytic Fungus Alternaria eureka 1E1BL1
The work seems to be of a decent quality for "Molecules" and therefore I would recommend this manuscript for publication.
Author Response
We would like to thank to the reviewer for very constructive comments.
Regarding the critique he/she raised ("Author needs to mention the incubation period with Endophytic Fungus Alternaria eureka 1E1BL1"), please note that incubation time is provided in the original abstracts as "After 21 days of incubation, ...".
Thanks again for positive comments and suggestions.
Reviewer 3 Report
Dear Authors,
This manuscript described the biotransformation of gitoxigenin by a fungus Alternaria eureka 1E1BL1 to generate five new cardenolide analogues. In addition, cytotoxic activity was tested, and found biotransformed metabolite 8 show promising cytotoxicity against A549, PANC-1 and MIA PaCa-2 cell lines, without exhibiting serious toxicity on healthy cell lines MRC-5 and HEK-293 at tested concentration.
1) At lines 36-38, "Microbial enzyme systems are able to catalyze regio- and stereo-specific reactions which are challenging issues in chemical synthesis", the cited reference is a bit old, Authors may consider to include below references.
a) J. Am. Chem. Soc. 2020, 142, 15, 7145–7152. DOI: 10.1021/jacs.0c01605
b) J. Am. Chem. Soc. 2021, 143, 1, 80–84. DOI: 10.1021/jacs.0c11226
c) J. Am. Chem. Soc. 2020, 142, 49, 20560–20565. DOI: 10.1021/jacs.0c10361
2) Figure 1, the front size is too small. And the L-oleandrose should be L-oleandrose.
3) Table 1, position 16, compound 1, δH 4.41, t, (16.5) is that correct? or it is 6.5 Hz?
4) Table 1, position 16, compound 4, how can δH 4.44 be a singlet? or it is a "broad singlet, brs"?
5) Table 1, position 17, compound 5, δH 3.29 d (8.0) has a coupling constant of 8.0 Hz and its coupling partner has δH 4.94 d 6.6 Hz. I think 8.0 Hz is a bit different from 6.6 Hz, and δH 4.94 should be a triplet. Authors should check again the 1H NMR.
6) I think Table 1 is too congested, maybe Author can consider use landscape orientation (mean 90 degree counter clockwise rotation of Table 1) for Table 1.
7) Figure 3, compound number is too small. Besides that, there is some serious flaw in Figure 3, compound 4, C-3 is a quaternary carbon, how can it has a HMBC correlation from H-3 to OMe units? I suggest Authors check carefully, and re-draw the HMBC arrows by prioritize and show all methyl HMBC correlations in the structure of Figure 3 and 4.
8) The assignment of opposite configuration at C-3 in 6 and 8 should discuss in detail.
Author Response
We would like to thank to the reviewer for very constructive comments and carefully going over the manuscript. We were able to most of the critiques that he/she raised. Please see our replies below.
Reply to critique 1:
The recommended references were replaced with the old ones.
Reply to critique 2:
Figure 1 was corrected as requested.
Reply to critiques 3 and 4:
Thanks to the reviewer for careful reading. The typo errors mentioned in critiques 3 & 4 were corrected.
Reply to critique 5:
Thanks to the reviewer for raising this important error. The spectrum was checked carefully. The H-16 resonance is overlapping with right foot of the water signal for compound 6 (not for 5), and it is difficult to deduce its coupling even with resolution boosting. Thus, H-16 resonance was reported as multiplet.
Reply to critique 6:
As requested, Table 1 was provided in landscape orientation.
Reply to critique 7:
For the first part, we would like to emphasize that HMBC correlations were shown from C atoms to hydrogen atoms, which was stated in the figures (arrows from C to H). It is more rational to draw the correlations from H to C due to nature of the HMBC experiment; however, we are in favor of keeping them in its original form due to simplicity. For the second part, the HMBC figures were re-drawn, and all methyl correlations were included.
Reply to critique 8:
We thank to the reviewer for raising this issue. The NOESY correlations proving the stereochemical alteration (epimer formation) at C-3 were mentioned in the text for clarification.
Round 2
Reviewer 3 Report
Dear Authors,
Thank you for addressing the comment positively.
Author Response
We would ilke to thank the reviewer for positive reply.